# Insights from In Vivo Studies of Cellular Senescence

**DOI:** 10.3390/cells9040954

**Published:** 2020-04-13

**Authors:** Luis I. Prieto, Sara I. Graves, Darren J. Baker

**Affiliations:** 1Department of Biochemistry and Molecular Biology, Mayo Clinic, Rochester, MN 55905, USA; prieto.luis@mayo.edu (L.I.P.); graves.sara@mayo.edu (S.I.G.); 2Department of Pediatric and Adolescent Medicine, Mayo Clinic, Rochester, MN 55905, USA

**Keywords:** senescence, senolytics, aging, mouse

## Abstract

Cellular senescence is the dynamic process of durable cell-cycle arrest. Senescent cells remain metabolically active and often acquire a distinctive bioactive secretory phenotype. Much of our molecular understanding in senescent cell biology comes from studies using mammalian cell lines exposed to stress or extended culture periods. While less well understood mechanistically, senescence in vivo is becoming appreciated for its numerous biological implications, both in the context of beneficial processes, such as development, tumor suppression, and wound healing, and in detrimental conditions, where senescent cell accumulation has been shown to contribute to aging and age-related diseases. Importantly, clearance of senescent cells, through either genetic or pharmacological means, has been shown to not only extend the healthspan of prematurely and naturally aged mice but also attenuate pathology in mouse models of chronic disease. These observations have prompted an investigation of how and why senescent cells accumulate with aging and have renewed exploration into the characteristics of cellular senescence in vivo. Here, we highlight our molecular understanding of the dynamics that lead to a cellular arrest and how various effectors may explain the consequences of senescence in tissues. Lastly, we discuss how exploitation of strategies to eliminate senescent cells or their effects may have clinical utility.

## 1. Introduction

Cellular senescence is a state of long-term exit from the cell cycle that can be induced in response to various forms of cellular damage. Much of the understanding of this complex state has come from experiments performed on cell lines exposed to various insults, including excessive oncogenic signaling, extreme DNA damage, and extended culturing time. In fact, it was nearly 60 years ago that Leonard Hayflick and Paul Moorhead first defined replicative senescence by finding that normal human fetal fibroblasts ceased proliferation after long-term passaging [1]. Very shortly thereafter, it was postulated that cells exhibiting this condition were responsible for tissue dysfunction associated with increasing chronological age. Indeed, the accumulation of senescent cells has been shown to correlate with age and disease, which may result from impaired clearance by the immune system [2]. However, conclusive evidence for causality behind a relationship between senescence and aging has remained elusive until very recently. Additionally, far less is known about senescence in vivo compared to what has been shown in vitro, as there have been very few ways to investigate this state. Simple issues, such as if this state is “permanent,” how exactly to best measure it, what stresses induce cells to become senescent in the first place, and why senescent cells persist instead of dying, remain unanswered in tissue contexts.

Several recent observations have reinvigorated the field to explore how senescent cells promote dysfunction in organisms. Introduction of a senescent-cell specific inducible suicide gene in mice has given us clues about the consequence of senescent cell accumulation in various tissues and diseases. In these animals, a portion of the promoter of a gene important for long-term cell cycle exit, p16^Ink4a^ (hereafter p16), is used to drive expression of this suicide construct (INK-ATTAC transgenic mice) and initiate apoptosis in a subset of senescent cells which express high amounts of p16 [3]. Elimination of p16-expressing senescent cells in these mice, beginning midlife, extended median lifespan, and attenuated a number of age-related deteriorations in tissue function [3]. Importantly, clearance of senescent cells can also ameliorate pathology in mice predisposed to chronic diseases [4,5]. While observations in INK-ATTAC mice have greatly extended our understanding of the consequences of senescent cell accumulation in vivo, these studies have caveats. Namely, this approach does not eliminate non-p16 expressing senescent cells, and not all p16-expressing cells are senescent (although no detrimental effects have been observed with long-term treatment of INK-ATTAC mice). In this perspective piece, we discuss the molecular dynamics that lead to cellular senescence and the consequences of senescent cell accumulation in vivo, paying particular attention to the role of p16 and p21^Cip1^ (hereafter p21) in these processes. We also discuss how we may be able to exploit these observations for generating pharmacological agents that can be used in clinical trials for patients.

## 2. Molecular Mechanisms of Cellular Senescence

A defining feature of cellular senescence is irreversible cell cycle exit. In response to a variety of intrinsic or extrinsic stresses, cells can engage the p53-p21 and/or p16-RB effector pathways to halt cell-cycle progression in an attempt to mitigate the damage that has occurred [6]. If the damage is irreparable, these cells can either die through apoptosis or survive by becoming robustly growth-arrested in the state of cellular senescence (Figure 1). In this way, senescence may act as a potent intrinsic tumor suppressor mechanism through upregulation of p53/p21 and p16. The ability of p53 to prevent the expansion of cells with potentially pre-neoplastic alterations has been elegantly described, as restoration of p53 expression in p53-null sarcomas induced senescence and suppressed tumorigenesis [7,8]. In accordance with these observations, the ablation of p53 in senescent human fibroblasts and mammary epithelial cells with low p16 expression allowed cell-cycle re-entry [9], which underscores the importance of p16 dynamics in cell-cycle arrest. Use of a transgenic mouse model that carries the whole p16 gene locus tagged with firefly luciferase demonstrated that the absence of p53 promotes p16 expression as a compensatory tumor suppressor [10]. While it has been established that p16 knockout mice are tumor prone, it remains poorly understood if p16 expression is required or sufficient to maintain stable cell cycle arrest. It will be crucial to further investigate the functions of p16 in cellular senescence in physiologically relevant settings by utilizing more sophisticated in vivo approaches such as cell-type or tissue specific overexpression or deletion of p16.

Studies in progeroid BubR1 hypomorphic (*BubR1*^H/H^) mice have demonstrated that the accumulation of senescent cells associated with elevated expression of p16, p19^Arf^, p53, and p21 contributes to tissue deterioration [11,12]. Germline inactivation of p16 attenuated the rate of senescent cell accumulation, whereas the ablation of p19^Arf^ further aggravated senescence and tissue deterioration in *BubR1*^H/H^ mice [12]. These results suggest that p16 is an effector and p19^Arf^ is an attenuator of cellular senescence in vivo. Furthermore, p21 expression, at least in BubR1 hypomorphic mice, appears to delay the onset of senescence and tissue dysfunction [11]. Although systemic genetic knockout mice are powerful tools, one caveat to this approach is that there may be compensation both during development and aging. It will be important to develop alternative approaches to overcome these limitations. Additionally, these tools will be able to answer some of the outstanding questions in vivo, such as whether senescence impacts specific cell populations, how the cross-talk between p53-p21 and p16 determines how cells commit to irreversible cellular arrest, and how the expression of p16 and p21 changes the fate and phenotype of a senescent cell. Importantly, understanding the function of these tumor suppressor genes in cellular senescence might provide clarity for classifying senescent cells, as there are likely multiple classes where some senescent cells may be beneficial, while others are pathological.

In addition to understanding what gene expression changes may induce senescence, we are beginning to appreciate how epigenetics play an important role in both maintaining and, potentially, promoting increased depth of senescence. For instance, chromatin regulation and lamin B1 reduction has been shown to enforce the stable cellular arrest underlying senescence [13,14]. Additionally, the changes in DNA methylation in a senescent cell have been shown to be similar to those in a cancer cell, suggesting that if senescent cells were to bypass the barrier of proliferation arrest, they may already have an epigenetic program that can promote tumor formation [15]. This may explain how senescence, an intrinsically tumor suppressive mechanism, is paradoxically capable of enhancing tumor formation [16]. Moreover, replicative senescence in human lung myofibroblasts has been linked with DNA hypomethylation and focal hypermethylation at CpG islands, which are features of genome instability and silencing of tumor suppressor genes [15,17]. However, in contrast to this, a recent study in human skin fibroblasts showed only minimal differences in DNA methylation in oncogene-induced senescence (OIS) versus replicative senescence. In order for tumor progression to occur, OIS cells would need to capable of escaping from senescence [18]. Interestingly, most of the DNA methylation changes seen in immortalized cells are independent of OIS and replicative senescence, suggesting that senescence-epigenetic changes likely do not contribute to the transformation of cells and senescence escape is unlikely [18]. However, it will be critical to know if the age-associated increase in cancer risk comes from transformation of senescent cells themselves or from paracrine effects of senescence that induce uncontrolled proliferation in neighboring cells. It has been reported that previously senescent cells that re-enter the cell cycle have a strong potential for tumor development [19]. Taken together, these studies indicate that epigenetic alterations may contribute to the dynamic process of cellular arrest and underlie further deepening of the phenotype.

## 3. Inducers of Cellular Senescence In Vivo

The identification of senescent cells is essential to determine their mechanistic contribution to aging and disease. The benefits of understanding how cells are induced to become senescent in vivo are two-fold: (1) understanding senescence induction in physiologically-relevant settings may provide new therapeutic insights for the prevention of senescent cell accumulation and (2) it will inform future experimental induction of senescence in studies aimed at dissecting the causative roles of senescence in aging and disease. Thus far, techniques to induce senescence are mostly employed in culture dishes and include serial passaging, irradiation, oxidative stress, and genotoxic stress. Less has been done to induce senescence in vivo, largely because these common methods of senescence induction in culture cannot be simply applied to a whole organism.

A causal role of senescence to tissue dysfunction was shown through the injection of senescent cells into young or aged mice, where there was evidence of dysfunction with aging [20]. Another study demonstrated obesity-induced senescence has the capacity to detrimentally impact neurogenesis and anxiety in mice [21]. While informative, both of these studies are limited in the context- and cell-type specificity of senescence. Importantly, there are a number of similarities in the expression of factors of the senescence-associated secretory phenotype (SASP) in a variety of different cell types, including fibroblast and epithelial cells, as well as in response to a variety of senescence inducing stimuli, including replicative senescence, OIS, and chemotherapy-induced senescence, although there are some factors that seem more specific for particular cells or stresses [22,23,24]. The SASP consists of a variety of bioactive molecules, including cytokines, chemokines, growth factors, and matrix metalloproteinases that are thought to disrupt the function of otherwise normal cells in the local environment (Figure 2) [23]. Senescent cells exert a majority of their negative effects by promoting chronic inflammation and tissue remodeling through the SASP. Therefore, differences in the SASP may determine what effects senescent cells may have on adjacent cells and it is necessary to increase our understanding of the cell- and context- specificity of senescence. For example, it was reported that OIS human diploid fibroblast cells have two recognizably different secretomes that are mediated by NOTCH1 [25]. Interestingly, in the first ‘wave’ of secretion, NOTCH1/TGFβ signal reinforces senescence in a positive feedback loop, while also inducing paracrine senescence in adjacent cells through the JAG1 ligand/NOTCH1 receptor interaction. The second ‘wave’ consists mainly of pro-inflammatory factors and metalloproteinases that might create an immunosuppressive microenvironment. Moreover, it has been shown that while oncogenic Ras induces the primary cell to senesce, senescence in adjacent cells, defined as ‘secondary (or bystander) senescence,’ is controlled by NOTCH [26]. Therefore, understanding more about how senescence is induced in vivo, i.e., what stresses in which contexts promote senescence, can lead us to accurately induce senescence in cell populations and contexts of interest when trying to study the effects of senescent cells on a tissue or organismal level. This section will describe what we currently know about senescence induction in vivo and discuss how we may utilize this knowledge to create new mammalian models for studying the causal roles of senescence in aging and disease.

### 3.1. Replicative Stress and DNA Damage

Cellular senescence was first described as an effect of prolonged culture and replicative stress [1] and then linked to telomere shortening as a consequence of the “end-replication problem.” Critically short telomeres trigger a DNA damage response that halts proliferation in cells until DNA is repaired. Intriguingly, DNA damage remains unresolved and the cell moves from a non-proliferative quiescence state to a robust, permanently growth-arrested senescence state. As such, inflicting DNA damage via irradiation or genotoxic drug treatment is a common method for inducing senescence in culture, as is culturing cells repetitively until critically short telomeres trigger this state of replicative senescence. Such in vitro studies have shed light on the molecular mechanisms of senescence, namely through activation of the p53-p21 pathway. However, how terminally-differentiated cells in an organism undergo senescence in the absence of shortening telomeres or unresolved DDR signaling is not yet clear.

Recently, studies have suggested that telomere-associated DNA damage can occur in senescent cells in vivo regardless of telomere length. Telomere-associated DNA damage foci accumulate with age, occur independently of telomere length, and are becoming a well-accepted biomarker for senescent cells in vivo [27,28,29]. In this way, telomerase-deficient mice with dysfunctional telomeres, such as the late-generation telomerase knockout mouse G3 terc^–/–^, may be useful models to understand how telomere-associated DNA damage may induce senescence in vivo and what downstream consequences exist. Some work has been done in this space implicating the loss of p21 in reversing the premature aging phenotypes seen in the G3 terc^–/–^ mouse [30]. In this way, inducing telomere-associated DNA damage, or at least the downstream p53/p21 signaling pathway, may be a useful and physiologically relevant way to induce senescence in vivo. Additionally, DNA damage induced by local radiation in organisms might be a suitable model to study tissue-specific senescence.

### 3.2. Oncogenic Stress

Paradoxically, excessive oncogenic signaling can lead to cell-cycle arrest. OIS, initially observed in mammalian cells in culture [31], has also been studied in vivo. Strong oncogene expression activates multiple pathways that lead to an accumulation of the tumor suppressor genes including p53 and p16 [31]. This phenomenon is seen by genetic overexpression of many oncogenic Ras isoforms, including Kras^G12V^, Kras^G12D^, Nras^G12D^, and Hras^G12V^ [32,33,34,35]. These activating mutations lead to both tumorigenesis and an increased burden of growth-arrested senescent cells. Importantly, it is now being appreciated that while a senescent cell does not form a tumor itself, it does have the capacity to amplify the proliferation of neighboring cells, likely through its SASP. Interestingly, the elimination of p16-positive cells extended cancer-free survival in naturally aged mice; however, it remains unclear if p16-positive cells induced by Ras oncogenic signals contribute to tumorigenesis [3]. Therefore, when studying OIS, it is important to consider these dual and contradictory effects of senescent cells in tumorigenesis. Interestingly, the idea of inducing OIS in a cell-type specific manner can allow us to investigate how the oncogene-induced pro-survival signals might change the internal cellular dynamics of senescent cells.

### 3.3. Inflammatory Signaling

Inflammation, in general, is an orchestrated process by a network of immune cells that respond to an injury or infection; the failure to resolve such can lead to many chronic diseases [36]. As we age, the adaptive immune system declines, but interestingly, the activity of the innate immune system increases, thereby promoting chronic inflammation. Claudio Francheschi first described this phenomenon and coined the term ‘inflammaging’ to describe it [37]. The acquisition of a pro-inflammatory SASP in senescent cells, which also accumulate with age, further propagate the inflammaging phenotype. Less appreciated is the fact that the pro-inflammatory environment of aged tissue can also induce cells to become senescent in a detrimental positive-feedback loop through a process called paracrine senescence [38]. In this way, models of chronic pro-inflammatory states may be useful for studying senescence in vivo. For instance, loss of the specific isoform, nfkb1, in mice induces a premature aging phenotype that depends on increased senescent cell burden and increased inflammation in multiple tissues [39]. Studies in this mouse have solidified the complex interaction between inflammation and senescence and offer one option for experimentally inducing cellular senescence in vivo. Inducing senescence by knockout of nfkb1 in a cell- or tissue-type specific manner using the cre/loxP system could offer a useful mechanism for studying inflammation-stress induced senescence in vivo in multiple cell types and disease contexts.

### 3.4. Metabolic Stress

Adipocyte progenitor cells can display markers of cellular senescence (such as p16, p19, p21, SA-β-Gal, and SASP) [3,40] in obesity and diabetes [3,41,42]. This has been associated with increases in oxidative stress and p53 expression [43]. In this way, feeding animals a high-fat diet is a simple and feasible in vivo model for examining how metabolic dysfunction may drive senescence and disrupt tissue homeostasis. Though highly beneficial in elucidating some of the mechanisms and effects of adipose tissue senescence, this high-fat diet model of senescence induction is limited in that it cannot be used to understand how senescence is induced in specific cell types or tissues with age. Related, oxidative stress alone has been suggested to have a major role in inducing senescence in vivo. Reactive oxygen species accumulate with aging and many age-related pathologies and treatment with anti-oxidants increase lifespan in rodents [44].

Other types of metabolic dysfunction, such as mitochondrial dysfunction, ER stress, and autophagy impairment have been implicated in senescence and discussed recently elsewhere [45,46,47,48]. Interestingly, it was found that knockdown of LSG1, a GTPase in charge of the biogenesis of the 60S ribosomal subunit, caused the perturbation of the endoplasmic reticulum homeostasis leading the cell to a senescent state [48]. Targeting some of these more systemic metabolic components may provide new avenues to study metabolic-stress-induced senescence in vivo. Doing this in a cell-type or tissue-specific manner, perhaps by utilization of available cre/loxP systems, may help narrow in on which types of senescent cells play the most important roles in driving aging and disease.

## 4. Cell Type Specificity of Cellular Senescence

Currently, it is unclear if all cell types are capable of entering a senescent state in vivo. The absence of a universal marker(s) for cellular senescence makes it complicated to define, as some cell types are post-mitotic through differentiation processes, such as neurons. Thus far, p16 and SA-β-Gal have been widely used as the main biomarkers to identify and isolate senescent cells [49,50]. However, despite most p16 and SA-β-Gal positive cells showing features reminiscent of cellular senescence, it remains unclear if all cells with high levels of these biomarkers are senescent [51,52]. For example, emerging evidence suggests that certain immune cell populations, such as macrophages, may display senescence-like features [16], which raises the question of whether these cells are or can become senescent. Additionally, the SASP shares many molecules that would be indicative of inflammation [53], adding to the difficult nature of defining whether immune cells can become truly senescent or merely reactive in aging and disease [29]. Moreover, from a mechanistic perspective, studies have suggested that cyclin-dependent kinases (CDKs) support the expression of proinflammatory genes through the activation of NF-κB and STAT3 [54]. Interestingly, transcription of proinflammatory genes occurs during the G1 phase of the cell-cycle, recruiting CDK6 to the nuclear chromatin, where it associates with NF-κB and STAT3 transcription factors. Surprisingly, the capacity of CDK6 to induce p16 expression and to recruit p65, an NF-κB subunit, is independent of its kinase function [54]. Thus, the upregulated expression of p16 in reactive cells may be a consequence of proinflammatory gene expression. For this reason, it will be important to address what functions CDKs have independently of the cell cycle and, importantly, what role CDK inhibitors, such as p16, have in this inflammatory context. This topic has become of great interest since most of the age-related chronic diseases, including cancer, involves an impaired immune system [55]. Altogether, this requires more investigation to clarify whether immune cells are being recruited and activated by senescent cells or if they themselves become senescent and drive tissue dysfunction with age.

Another key hallmark of cellular senescence is a ‘deregulated’ metabolism, which may provide useful in identifying senescent cells in vivo if senescent cells indeed have distinct metabolic signatures [56]. It has been reported that after DNA damage, expression of p21 in senescent cells induces mitochondrial dysfunction and increases the production of reactive oxygen species (ROS), which amplifies the DNA damage response in a positive feedback loop [57]. Additionally, alterations in mitochondrial abundance have been shown in senescent cells, as has mitochondrial regulation of the SASP [58,59]. Therefore, a plausible alternative to identify senescent cells in vivo is by using a novel metabolic stable-isotope tracer, which localizes to the inner membrane of the mitochondria providing a radioactive signal that can be observed with positron emission tomography (PET) [60]. This approach can be used in conjunction with transgenic mouse models to visualize which cells are prone to senesce with age and age-related diseases and, importantly, to profile their metabolic signatures, such as oxidative phosphorylation. This strategy might provide a different perspective to enhance our ability to detect and quantify senescent cells and understand the differences in metabolic changes that occur in cells with age, and what risk they impose.

## 5. Senolytics

Recent observations using genetically-modified animal models indicate that the elimination of senescent cells attenuates aging and age-related diseases [3,4,5]. These findings have opened new avenues to explore pharmacological approaches to induce apoptosis in senescent cells termed senolytics.

With the absence of a unique marker for senescence, interventions have been developed to take advantage of some vulnerabilities that senescent cells have. Senescent cells, like cancer cells, are resistant to apoptosis through the upregulation of BCL-2 anti-apoptotic proteins. Efforts in cancer research have found ABT263 (navitoclax), a potent inhibitor of BCL-2 and BCL-xL anti-apoptotic proteins, can be used to treat lymphomas and other types of cancer [61]. Interestingly, due to the overexpression of BCL-2 in senescent cells, ABT263 exhibits senolytic activity and prolongs healthy lifespan in normally-aged mice [62]. Importantly, using mouse models of age-related chronic diseases, including atherosclerosis and neurodegeneration, in which the accumulation of senescent cells is detrimental, treatment with ABT263 attenuated disease pathology similarly to what was observed with genetic approaches for senescent cell ablation [4,5]. More in-depth, these approaches were tested in the early stages of the disease to investigate if senescent cells contribute to the development of the pathology. Thus, a subsequent investigation is required to directly assess the impact of eliminating senescent cells at later stages of established pathology. Additionally, the efficacy of ABT263 as a senolytic in cancer treatment still needs to be determined. Since senescent cells can intrinsically protect from tumor development but also amplify tumorigenesis, it will be interesting to determine the outcome of eliminating senescent cells in mouse models of tumorigenesis. Unfortunately, side effects can also be seen upon ABT263 treatment, namely the reduction of neutrophils (neutropenia) and thrombocytes (thrombocytopenia), as these cell types are also reliant on BCL-2 for their survival. Therefore, precautions should take place to alleviate these known side effects with intermittent administration to target senescent cells safely.

Additional pharmacological interventions have been shown to target senescent cells in mice. These include a peptide that disrupts the interaction of FOXO4 and p53 leading to apoptosis [63], as well as nanoparticles that target senescence-associated β-gal positive cells [64]. Interestingly, using a mouse model for osteoarthritis (OA), it was shown that senescent cells accumulate in articular cartilage and synovium promoting the development of OA. Consistent with the concept that senescent cells drive pathology, local administration of a new molecule named UBX0101, was used to disrupt the interaction of MDM2 and p53 to trigger apoptosis in senescent cells, yielding positive results in attenuating OA, plus validating UBX0101 as a senolytic [65]. Consequently, UBX0101 was initiated in a clinical trial with adult patients diagnosed with femorotibial osteoarthritis to evaluate the safety, tolerability, and pharmacokinetics of the drug. It is currently being evaluated in phase 2 clinical trials for the effectiveness in treating musculoskeletal diseases with an emphasis on patients with OA.

Natural compounds, such as quercetin and fisetin, have also been used in combination with anti-cancer drugs, particularly the pan-tyrosine kinase inhibitor dasatinib to treat naturally aged mice [20] and senescence-related diseases [66]. The combination of dasatinib and quercetin (D + Q) target particular sensitivities of pro-survival pathways found in senescent cells, known as senescent cell anti-apoptotic pathways (SCAPs) [67]. These drugs could theoretically influence a broad spectrum of pathways in all cells, which may make it difficult to assess if senescence ablation occurs or there has been some amelioration of key features of senescent cells, such as the SASP. Nevertheless, these findings set a foundation to start clinical trials in adult patients with idiopathic pulmonary fibrosis [68] and diabetic kidney disease [69]. These studies have provided evidence that intermittent doses of senolytics can be systematically used in humans and tolerated.

As the field of senescence is evolving, we are appreciating the importance of the context where senescent cells emerge. Thus, the focus should shift to create ‘targeted’ senolytics. In other words, instead of using drugs that act in a myriad of cells that could lead to side effects, developing specific targeted approaches will allow us to concentrate on key cells responsible for the dysfunction. For instance, OIS cells have different properties compared to replicative senescent cells [22]. The original cell type that gives rise to the senescent cell (i.e., pre-adipocytes; glial cells) also seems to matter in this context. Having this knowledge might give us the ability to design and administer drugs that act in specific senescent cell populations without further consequences.

The objective is to start treating chronic diseases from the root and not the symptoms of the disease. Unfortunately, most chronic diseases are complex, where not only one, but several cell-types contribute to the malady. As we are starting to enroll patients in “senolytics-clinical trials,” it will be imperative to assess if senolysis efficiently targets the primary cause of disease or if it works best in combination with other drugs. Therefore, since this might be context-dependent, additional basic science research is required to address the fundamental role of senescent cells, especially in the established contexts of disease.

## 6. Conclusions

The current philosophy of the healthcare system is to systematically treat chronic diseases with medications that, for the most part, address the consequence rather than the cause of the malady. Our desire to improve human care has motivated us to investigate how and why senescent cells accumulate with age and whether they may play therapeutically-relevant casual roles in age-related diseases. As we have begun to appreciate, senescence is not a static, but a dynamic process that is continuously evolving to halt the uncontrolled proliferation of a cell. However, despite their innate tumor suppressive effects, chronic accumulation of senescent cells drives impaired tissue function. Going forward, it is imperative to investigate the mechanisms that induce, maintain, and drive the detrimental effects of cellular senescence, specifically in different contexts in vivo. We have discussed several promising strategies that may be employed to better define how senescent cells drive dysfunction at the greater tissue and organismal levels. Elimination of senescent cells attenuates age-related diseases; however, given the dual and contradictory roles of senescent cells, future studies are necessary to provide a more concise biological understanding of the specific molecular dynamics underlying cellular senescence. These studies will form the essential basis for establishing new senolytic strategies to prevent or delay tissue dysfunction and increase healthspan.

## Figures and Tables

**Figure 1 cells-09-00954-f001:**
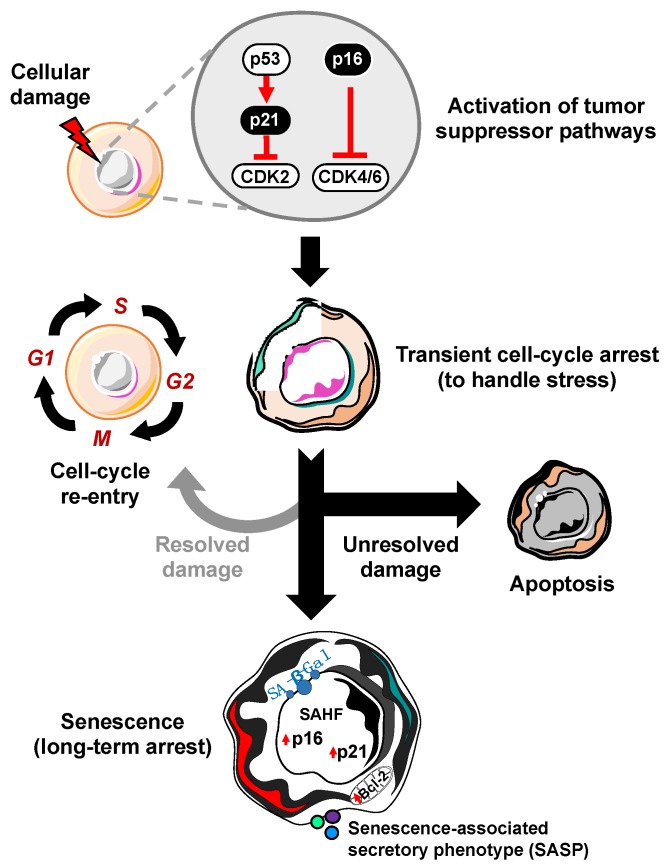
Process of cellular senescence. In response to cellular damage, the cell activates a myriad of pathways including the p53-p21 and p16-RB tumor suppressor pathways for transient cell-cycle arrest to handle and repair the stress. If the damage is repaired, it can re-enter the cell-cycle. However, if the damage is unresolved, the cell has two main fates: apoptosis (cell death) or cellular senescence, characterized by increased expression of cyclin-dependent kinase inhibitors (i.e., p16, p21), senescence-associated heterochromatin foci (SAHF), Bcl-2 anti-apoptotic proteins, senescence-associated beta-galactosidase (SA-β-Gal), and the senescence-associated secretory phenotype (SASP).

**Figure 2 cells-09-00954-f002:**
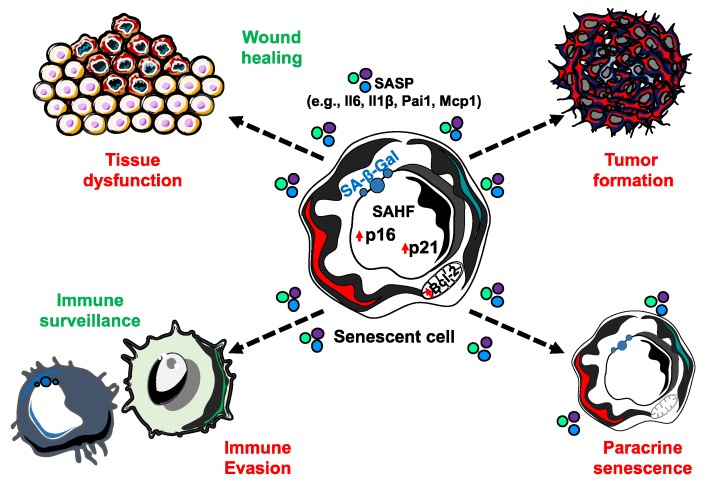
The senescence-associated secretory phenotype (SASP) is able to elicit a number of biological effects. While senescent cells are incapable of proliferation, they remain metabolically active and capable of secreting an array of bioactive molecules that have extrinsic impacts. SASP factors can function both in beneficial (green) and detrimental (red) biological processes.

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
