# Peer review of "Insights from In Vivo Studies of Cellular Senescence"

_cells, 2020, doi:10.3390/cells9040954_

Round 1
Reviewer 1 Report
This is a very interesting review article from the group whose contributions have appeared in in a number of major publications by targeting p16 high senescent cells with wide ranges of implications in mice. It will therefore timely be important to summarize and discuss their findings all together, the implications to humans, prospective in future directions, and more importantly the mechanisms to validate and call for more collaborative approaches across different fields. The review should also serve well for proper understandings of general readers and experts in different disciplines. Therefore, the review should focus on p16 related mechanisms in the types of senescence, responses and locations of senescent cells such as immune responses and metabolic responses. Some specific comments are the following:
- Authors describe in their first sentence that cell senescence is SASP. This is misleading and reflecting that authors lack understanding of the nature of cell senescence.
- Worse is the next sentence stating that much of what is known is derived from in vitro studies, implicating SASP is from in vitro studies.
- In the 5th line from the bottom in the abstract, in vivo biology is wrong because authors would not apparently like to define biology in vivo and in vitro
- Page 3 line 113, typo to correct “induction of senescence induction”
- Lines 111-120, discussions on references 13-15 are superficial and inappropriate, potentially misleading in non-expert readers. So, in-depth discussions are needed
- Lines 136-137: Please expand this sentence “The importance of this comes from studies that show differences in the senescence-associated secretory phenotype (SASP) from replication-induced versus OIS cells.” Are you implicating a significant difference of SASP between different types of senescence? These deserve citing and explaining.
- Lines 193-195, it would be very relevant to discuss if OIS involves increased p16, and if targeting p16-positive senescent cells might contribute to tumorigenesis
- Lines 202-216, the relationship between SASP and inflammaging deserves more discussions especially on the front of the mechanisms of SASP. If senescent cells secrete cytokines and inflammatory factors, what are the differences then between senescent cells and immune cells? How senescent cells can express and secrete proteins normally from innate immune cells. What gene expression programs are activated in senescence and why? Given targeting p16 high cells improves SASP and immunity, is that possible that senescent cell provides antigens to immune cells for immune cells to secrete cytokines in situ, and removal of senescent cells leads to removal of the antigens associated with aging thus SASP?
- Lines 218-226, while it seems clear that targeting p16 or p53 has impact on obesity, it is hard to imagine adipocyte senescence. Therefore, it is paramount to introduce the concept and mechanisms of how adipocyte as a terminally differentiated cell undergoes or behaviors or manifests senescence. Is senescence. Is senescence defined by SASP?
- Lines 241-246: These sentences are difficult to understand and serve better explanations: “Seminal studies have suggested that cyclin-dependent kinases (CDKs) support the expression of proinflammatory genes through the activation of NF-B and STAT3 [40]. Transcription of proinflammatory genes occurs during the G1 phase of the cell-cycle recruiting CDK6 to the nuclear chromatin where it associates with NF-B and STAT3 transcription factors. Interestingly, the capacity of CDK6 to induce p16 expression and to recruit p65, an NF-B subunit, is independent of its kinase function [40].”
Reviewer 2 Report
In this review, the authors summarise the main effectors of senescence, suggesting suitable in vivo models for the study of senescence in an organismal context. This is a well written review which touches on a lot of aspects of senescence and is therefore quite dense, taking a bird eye overview of the field. I only have a few minor additions that, I believe, would extent some points of the manuscript further.
Line 146: Mention Notch induced primary senescence (Hoare et al 2016 Nature Cell Biology) and Notch secondary senescence (Teo et al 2019 Cell Reports) here as these provide insights into effects on adjacent cells and heterogeneity between primary and secondary senescence
Section Replicative stress and DNA damage
How about local radiation induced damage in vivo as a suitable model to study effects of DNA damage mediated senescence?
Metabolic stress: comment on and cite Pantazi et al 2019 Aging Cell, ribosome biogenesis as a further mechanism of senescence induction in the context of metabolic dysfunction
Reviewer 3 Report
This review is meant to cover in vivo senescence and the therapeutic (including clinical) utility of senolytics, which is a very relevant and timely topic. The authors are leaders in the field of senescence and senolytic approaches and their work is undoubtedly outstanding.
However, in this review it seems that the authors tried to cover too many aspects of the cellular senescence, and because of the short length of the review, this was done too superficially, ending with more questions than answers. Key points such as cellular specificity of in vivo senescence and timing for senolytic treatment could be better addressed.
The review would be greatly improved by:
1- Removing or simplifying section 2 (what need to be said is in the legend of the figure 1), limiting the in vitro data and focusing more on in vivo senescence. I would delete the paragraph on epigenetics.
2- Sections 4 and 5 are the most interesting. Indeed, when dealing with in vivo senescence, the key question is, in a pathological context which cell type becomes senescent and contributes to the disease? Accordingly, targeting this specific cell type with a senolytic is the main goal.
The review should therefore focus more on 2 major points that are only shortly mentioned:
1) Can we detect and quantify senescent cells in different tissues and cell subpopulations in vivo?
2) When senolytics should be used? In primary treatments, that can only be performed in animal models, or in secondary treatments as in clinical trials? This important question is only mentioned in the conclusion, but it should be integrated throughout the review, particularly in the Section 5 on senolytics; it would be important to clarify at what stage of the pathological process each senolytic approach was tested. This would help to understand the clinical relevance of the senolytics.
Globally, the review should give more answers than raise questions about in vivo senescence.
Round 2
Reviewer 1 Report
Authors revision is much appreciated. Just a couple of specific points here for authors to consider:
Line 79-81: Here please consider reviewing what evidence and if it is enough to establish the notion that senescence is tumor suppressive and if senescent cells can actually spark off oncogenesis. Is the current progress allowed such a clear cut statement that senescence is tumor suppressive? This is particularly important as tumorigenic process takes place in association with aging, so senescence may not be necessarily tumor suppressive as once thought but instead oncogenic. This is perhaps also supported by authors’s findings that inhibition of senescence like phenotype is inhibitory to oncogenesis.
In line 326-327: I would not think that the adipocyte progenitor cells are independent of or more important than inflammatory cells in obesity and diabetes; and in line 350-352: Given that emerging evidence suggests that certain immune cell populations, such as macrophages may display senescence-like features [16], it would important to refer the following three papers and discuss a potential involvement of inflammation in senescence. 1-3 In Ref. No. 1 authors actually suggest that SASP is originated from inflammatory cells rather than senescent cells. So is it possible and if yes to what degree the in vivo studies on cell senescence are actually on inflammation.
1 Chen, R. et al. Telomerase deficiency causes alveolar stem cell senescence-associated low-grade inflammation in lungs. The Journal of biological chemistry 290, 30813-30829, doi:10.1074/jbc.M115.681619 (2015).
2 Jurk, D. et al. Chronic inflammation induces telomere dysfunction and accelerates ageing in mice. Nature communications 2, 4172, doi:10.1038/ncomms5172 (2014).
3 Yan, J. et al. Obesity- and aging-induced excess of central transforming growth factor-beta potentiates diabetic development via an RNA stress response. Nature medicine 20, 1001-1008, doi:10.1038/nm.3616 (2014).
Author Response
Line 79-81.
We have modified the text to incorporate this comment.
Line 326-327
We have reworded this section to prevent this interpretation that adipocyte progenitor cell senescence is independent or more important. We simply state the evidence that these cells exhibit characteristics of senescence in these conditions.
Line 350-352
As suggested, we have added the first two suggested references. We thought about the best way to incorporate the third suggested references, but it felt out of place and inadequately discussed. Therefore, we have not included this specific reference in this revised version. We apologize for being unable to accommodate this request in the current format.
Reviewer 3 Report
No more comments.
Great improvements from the previous version.
Author Response
We thank the reviewer for the positive comments in response to our previous submission.